# Facile Synthesis of Nitrogen-Doped Carbon-Supported Rhodium–Cobalt Alloy Electrocatalyst for Oxygen Reduction Reaction

Sujung Park [1,†], NaHyun Park [1,†], Muthuchamy Nallal [1,2], Mohammad Yusuf [1], Sungkyun Park [3], Jae-Myung Lee [4,*] and Kang Hyun Park [1,*]

1 Department of Chemistry, Pusan National University, Busan 46241, Korea
2 Department of Physics and Institute of High Pressure, Hanyang University, Seoul 04763, Korea
3 Department of Physics, Pusan National University, Busan 46241, Korea
4 Department of Naval Architecture and Ocean Engineering, Pusan National University, Busan 46241, Korea
* Correspondence: jaemlee@pusan.ac.kr (J.-M.L.); chemistry@pusan.ac.kr (K.H.P.);
  Tel.: +82-51-510-2238 (K.H.P.)
† These authors contributed equally to this work.

**Abstract:** Fuel cells are considered as efficient and environmentally ecofriendly alternatives for energy production. The oxygen-reduction reaction is important in energy-conversion systems for fuel cells. In this work, rhodium (Rh) and cobalt (Co) alloy nanoparticles were deposited on nitrogen (N)-doped carbon (C) supports (RhCo/NC) using ball milling and thermal decomposition. The RhCo/NC composites were transformed into small nanoparticles with an average diameter of approximately 4 nm. The properties of the as-synthesized RhCo/NC nanocatalyst were characterized through transmission electron microscopy, X-ray diffraction, X-ray photoelectron spectroscopy, and Raman spectroscopy. The catalytic activity of the nanocatalyst for the ORR was investigated. The RhCo/NC nanocatalyst showed good activity for the ORR, long-term durability in chronoamperometry tests, and resistance to methanol crossover in an alkaline solution. This was because of the synergistic effects of the metal alloy. Chronoamperometric analysis demonstrated the remarkable durability of the RhCo/NC nanocatalyst compared to a commercial platinum (Pt)/C catalyst. Moreover, the RhCo/NC nanocatalyst exhibited good methanol tolerance. The RhCo/NC nanocatalyst can replace Pt-based catalysts in energy-conversion systems.

**Keywords:** rhodium-cobalt alloy; nitrogen-doped carbon; electrocatalyst; oxygen reduction reaction

## 1. Introduction

The increase in energy consumption and the problems caused by toxic air pollutants have led to the development of renewable and sustainable energy systems and green-energy conversion and storage devices, such as fuel cells (proton exchange membrane fuel cells, direct methanol fuel cells, etc.) and metal–air batteries [1–4]. In particular, fuel cells are considered to be ecofriendly because only water and waste heat are released and chemical energy is converted to electrical energy with high efficiency. Catalysts are important components of the cathodes of fuel cells [2,5–7]. This is because the oxygen-reduction reaction (ORR), which is a sluggish kinetic reaction, is the rate-determining step [8,9].

Platinum (Pt) is the most suitable electrocatalytic catalyst, particularly for the ORR at the cathodes of fuel cells [10,11]. However, it has intrinsic limitations such as high cost, scarcity, low durability, and weak methanol tolerance for the ORR. These have severely restricted its industrial applications. To solve these issues, numerous attempts have been made to design and construct the Pt group electrocatalysts by slightly modifying the size, shape, structure, and composition [12,13]. One of the most successful strategies is the fabrication of alloy nanocatalysts with precious metals (ruthenium, palladium, silver, rhodium (Rh), etc.) [12–16] or non-noble metals (cobalt (Co), iron, manganese, nickel, etc.) [17–22].

Rh is a promising catalyst for the ORR because of its diversity, high activity, and selectivity for numerous electrochemical reactions [23,24]. In addition, Rh-based electrocatalysts show efficiently enhanced methanol tolerance capacity towards ORR performance [25,26]. Transition metals exhibit significantly high ORR performance; however, their practical applications are limited by their relatively low surface areas and conductivity. These issues can be solved by integrating metals with a nitrogen (N)-doped carbon (C) matrix [27,28].

In this paper, we report the facile preparation of N-doped C-supported Rh–Co alloy (RhCo/NC) nanoparticles using high-energy ball milling and thermal treatment with a solid mixture of metal salts and urea under nitrogen gas ($N_2$) flow (Scheme 1). Ball milling is widely used to synthesize heteroatom-doped C supports. It is a well-known technique for mechanically producing uniform-sized particles. Thermal treatment is used to allow the efficient surface area of composites [7,22,29]. The obtained uniformly dispersed RhCo/NC nanocatalyst is utilized as an electrocatalyst in the ORR.

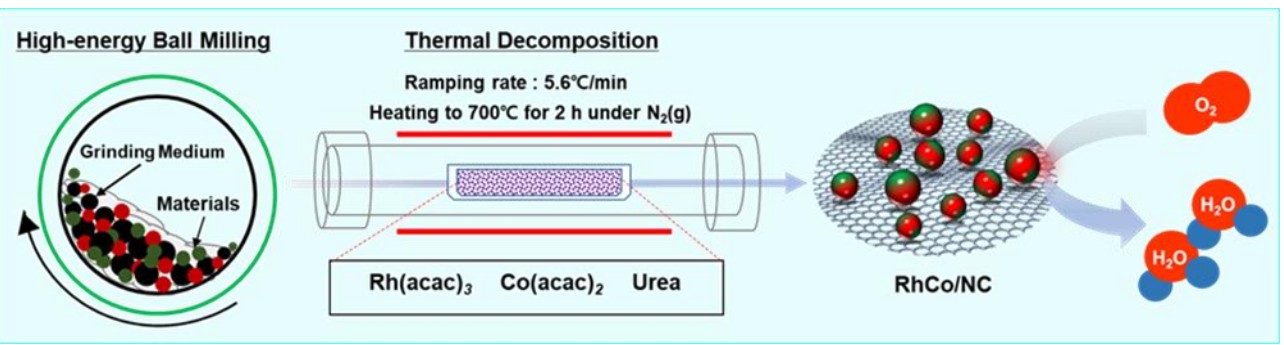

**Scheme 1.** The schematic diagram of nanomaterials, RhCo/NC fabrication, and application toward ORR.

## 2. Experimental Methods and Measurements

### 2.1. Chemicals

Rhodium (III) acetylacetonate (Rh(acac)$_3$, Rh(C$_5$H$_7$O$_2$)$_3$, 97%), cobalt (II) acetylacetonate (Co(acac)$_2$, Co(C$_5$H$_7$O$_2$)$_2$, 97%), urea (CH$_4$N$_2$O, 99.0–100.5%), potassium hydroxide (KOH, 99.99%), and Nafion were obtained from Sigma-Aldrich. 2-propanol ((CH$_3$)$_2$CHOH, 99.5%) was obtained from Alfa Aesar.

### 2.2. Preparation of RhCo/NC

First, Rh(acac)$_3$ (0.16 g) and Co(acac)$_2$ (0.10 g) in a molar ratio of 1:1 and urea (1.25 g) were added to a ball-mill vial. Then, the powder mixture was high-energy ball-milled at 1725 rpm for 5 min. The mixed powder was placed in a tube furnace, and samples were prepared by increasing the temperature to 700 °C at 5.6 °C·min$^{-1}$ under $N_2$ flow (200 mL·min$^{-1}$). The resulting catalyst was cooled to room temperature and collected (Scheme 1). The detailed characterizations are described in the supporting information (SI-1).

### 2.3. Electrocatalytic Measurements

The electrochemical performance of the RhCo/NC catalyst for the ORR was evaluated using a potentiostat (ZIVE SP1, Zivelab) and a modulated speed rotator (AFMSRCE, Pine) at room temperature. A saturated calomel electrode (SCE; sat. sodium chloride) was used as the reference electrode and a graphite electrode as the counter electrode in a three-electrode configuration. A glassy carbon rotating disk electrode (RDE, diameter: 5 mm) was used as the working electrode. All potentials were transformed from the SCE scale to the reversible hydrogen electrode (RHE) scale using $E_{RHE} = E_{SCE} + 0.0592 \times pH + E_{SCE}^{\circ}$ in a 0.1 M potassium hydroxide (KOH) alkaline electrolyte. The RhCo/NC catalyst and a commercial 20 wt% Pt/C catalyst were dispersed for 1 h in 2-propanol (200 μL), deionized water (200 μL), and a 5% Nafion solution (20 μL) to prepare homogeneous inks by sonically mixing 5 mg of the catalysts. In total, 7 μL of the nanocatalyst ink was loaded on the

surface of the RDE and dried at 60 °C for 1 h. The ORR performances were measured in 0.1 M KOH alkaline electrolyte after purging with $N_2$ or oxygen gas ($O_2$) for 30 min. Cyclic voltammetry (CV) was performed at 0.05–1.22 V for 30 cycles at a scan rate of 10 mV s$^{-1}$. Linear sweep voltammetry (LSV) was conducted at a scan rate of 5 mV s$^{-1}$ and rotation speeds of 400–2400 rpm. The chronoamperometric performance was measured at 0.56 V and a rotation speed of 1600 rpm to examine the long-term durability. In addition, methanol tolerance was investigated by adding 1.0 M methanol to the 0.1 M KOH aqueous solution after 500 s. The electron transfer number (n) was calculated from the Koutecky–Levich (K–L) plot, which follows Equations (1)–(3).

$$\frac{1}{J} = \frac{1}{J_L} + \frac{1}{J_K} = \frac{1}{B\omega^{\frac{1}{2}}} + \frac{1}{J_K} \tag{1}$$

where J, $J_L$, and $J_K$ are the current density, diffusion-limiting current density, and kinetic current density, respectively; $\omega$ is disk rotation speed ($\omega = 2\pi N$, N is the linear rotation speed); and B is given by Equation (2).

$$B = 0.62nFC_O(D_O)^{2/3}\vartheta^{-1/6} \tag{2}$$

$$J_K = nFkC_0 \tag{3}$$

where F is the Faraday constant (96, 485 C mol$^{-1}$), $C_O$ is the bulk oxygen concentration ($1.2 \times 10^{-6}$ mol cm$^{-3}$), and $D_O$ is the oxygen diffusion coefficient in the bulk solution ($1.9 \times 10^{-5}$ cm$^2$ s$^{-1}$). $\vartheta$ is the kinematic viscosity of the electrolyte, and k is the electron transfer rate constant. Then, n is determined from the slope of the $J^{-1}$ vs. $\omega^{-1/2}$ plot.

## 3. Results and Discussion

### 3.1. Characterization of RhCo/NC Nanocatalyst

Scheme 1 illustrates the procedure for the synthesis of the RhCo/NC nanocatalyst. We fabricated RhCo/NC nanoparticles via a facile synthesis method that involved high-energy ball milling and thermal decomposition. First, Rh(acac)$_3$ and Co(acac)$_2$ as metal precursors and urea as the N and C source were mixed in the solid state. Subsequently, the thermal decomposition of the solid mixture was carried out at 700 °C under $N_2$ flow. At high reaction temperatures, the Rh and Co precursors were annealed and reduced to form Rh–Co alloy nanoparticles. Urea was simultaneously transformed into a graphitic C support, with the release of ammonia and carbon dioxide. We used significantly lower amounts of metals and successfully synthesized a size-controlled catalyst. The morphology of the RhCo/NC composites was examined using transmission electron microscopy (TEM) images. Figure 1a shows the well-dispersed small nanoparticles on the C nanosheets. As shown in Figure S1, the average size of the Rh–Co alloy nanoparticles was 3.59 nm, which was determined by measuring 400 particles. Such a particle size may imply the high electrocatalytic activity of the fabricated RhCo/NC catalyst for the ORR.

The high-angle annular dark-field scanning TEM image (Figure 1b) showed white dots, which indicated metal-rich regions in the C support. Furthermore, Rh (Figure 1c) and Co (Figure 1d) elementary mappings showed that the nanoparticles were located on the C support. Figure 1e,f shows the high-resolution TEM (HR–TEM) image and its Fourier transform image to examine the morphology of the catalyst in detail. The HR–TEM image showed that the lattice crystalline fringes of RhCo/NC corresponded to the (111) planes of the Rh–Co alloy with an interplanar distance of 0.212 nm. In addition, energy-dispersive X-ray spectroscopy and combustion analysis showed that the RhCo/NC nanocatalyst contained 49.8% Rh and 50.2% Co. Thus, the molar ratio of Rh and Co was demonstrated to be 1:1, which was similar to the experimental results. As a result, we successfully developed a size-controlled nanocatalyst with uniform dispersion.

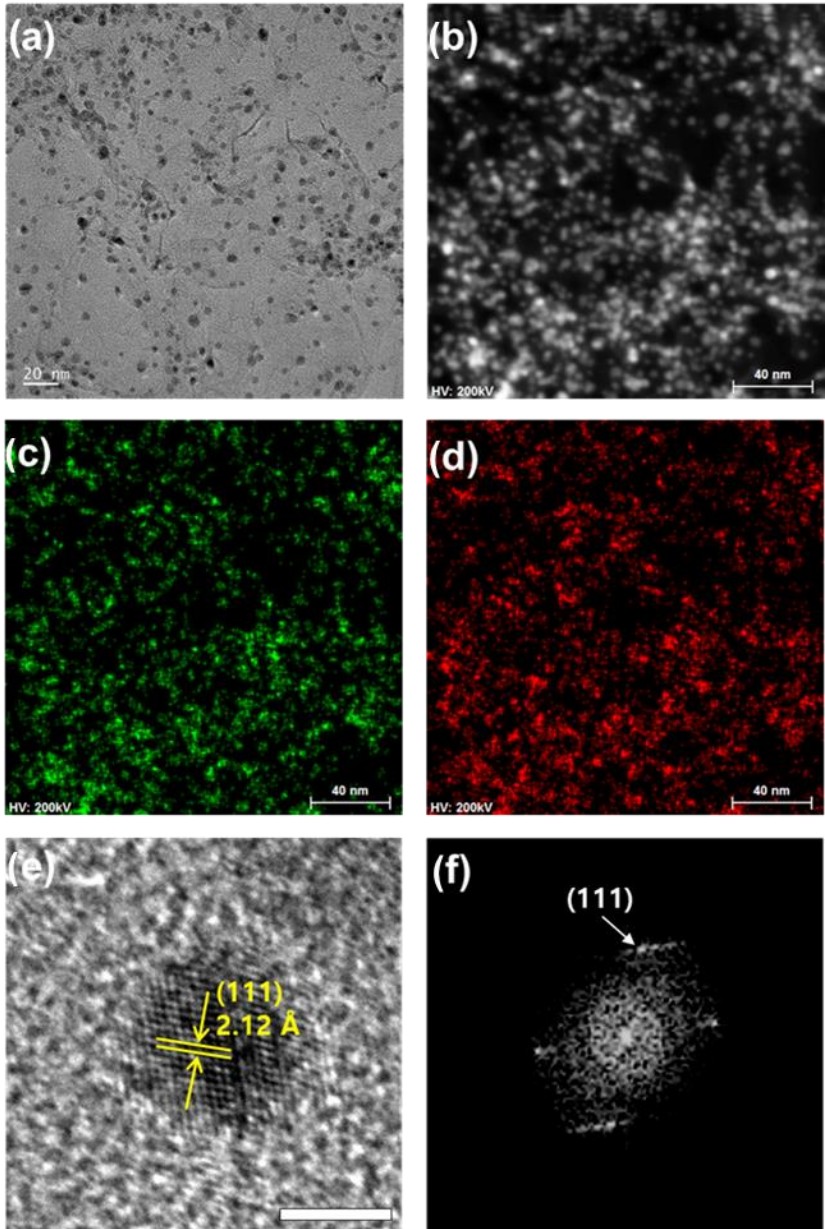

**Figure 1.** (**a**) TEM image and (**b**) HAADF–TEM image of RhCo/NC catalyst; (**c,d**) elemental (Rh, Co) mapping images. (**e**) HR–TEM image of RhCo/NC catalyst with (111) lattice space (bars express 4 nm), and (**f**) FT image of the TEM image of (**e**).

The crystalline structure of the RhCo/NC catalyst was examined by indexing the X-ray diffraction (XRD) spectrum to the face-centered cubic CoRh phase according to JCPDS No. 01-071-0721 (Figure S2). The crystalline size of the RhCo/NC catalyst calculated using Scherrer's equation for the diffraction peak of the (111) plane was 4.3 nm, which was similar to the particle size obtained using the TEM images. Raman spectroscopy was performed to study the crystallinity of the as-prepared C materials. The Raman spectrum of the fabricated RhCo/NC composites is shown in Figure S3. The Raman spectrum exhibited three typical bands, i.e., the G, D, and 2D bands (1584 cm$^{-1}$, 1357 cm$^{-1}$, and 2720 cm$^{-1}$). $I_G/I_D$ was calculated as 0.92, demonstrating the high graphitization of C and low crystallinity of the C support. In addition, the second-order 2D band was a characteristic of graphitic sp2 materials. X-ray photoelectron spectroscopy (XPS) was used to examine the chemical states and elemental compositions of the RhCo/NC nanocatalyst (Figure S4). The peaks detected in the XPS spectra were attributed to Co, O, N, and Rh (Figure S4a). As shown in

Figure S4b,c, the Rh 3d and Co 2p peaks of Rh–Co alloy nanoparticles were doublet peaks at 307.1 eV and 311.9 eV, which were assigned to Rh $3d_{5/2}$ and Rh $3d_{3/2}$ of the $Rh^0$ signal, respectively. Figure S4d shows the N 1s peaks of the RhCo/NC catalyst. Pyridinic N and pyrrolic N were observed at 398.2 eV and 400.8 eV, respectively, which were related to the type of the N atoms in the N-doped C support. In particular, pyridinic N increased the ORR activity. The N content in the RhCo/NC catalyst was determined as 4.2 wt.%.

### 3.2. Electrocatalytic Performance in ORR

The RDE performance of the fabricated nanocatalyst was used to examine the ORR activity. Figure 2a shows the CV results for the RhCo/NC nanocatalyst. The intrinsic oxidation/reduction reaction was conducted in N-saturated and O-saturated 0.1 M KOH alkaline solutions at a scan rate of 20 mV $s^{-1}$ in the potential window. Notably, the specific cathodic peak at 0.81 V vs. the RHE was attributed to the reduction in molecular oxygen. However, cathodic peaks were not observed in the N-saturated electrolyte. The electrocatalytic activity for the ORR was demonstrated by performing LSV measurements in O-saturated 0.1 M KOH at a scan rate of 5 mV $s^{-1}$ and a rotation speed of 1600 rpm. Figure 2b shows a comparison of the ORR activity of the RhCo/NC nanocatalyst and commercial 20 wt% Pt/C catalyst. The RhCo/NC catalyst showed remarkable catalytic activity with an onset potential ($E_{onset}$) of 0.95 V vs. the RHE and a half-wave potential ($E_{1/2}$) of 0.84 V vs. the RHE, which approached the catalytic activity of the Pt/C catalyst ($E_{onset}$ and $E_1$ were 1.02 V and 0.89 V vs. the RHE, respectively). The comparison of the LSV results for the RhCo/NC and Pt/C catalysts showed that the Pt/C catalyst exhibited a better positive onset potential and diffusion-limited current density ($j_L$). However, it should be noted that the difference between the values of $E_{onset}$ for both catalysts was only 0.07 V and the values of $j_L$ were quite close. Figure S5 shows the Tafel plots obtained at a rotation speed of 1600 rpm using the LSV polarization curves. The Tafel slopes for the Pt/C and RhCo/NC catalysts were 97.32 mV $dec^{-1}$ and 82.22 mV $dec^{-1}$, respectively. These results showed that the fabricated RhCo/NC nanocatalyst had a faster electron transfer rate in the ORR compared to the Pt/C catalyst. This implied that the C support and heteroatom doping increased the electron transfer rate and active sites during the ORR. Notably, the loading mass of Rh–Co in the RhCo/NC nanocatalyst was half of that of Pt in the Pt/C catalyst. Thus, the RhCo/NC nanocatalyst was more cost-effective than the Pt/C catalyst.

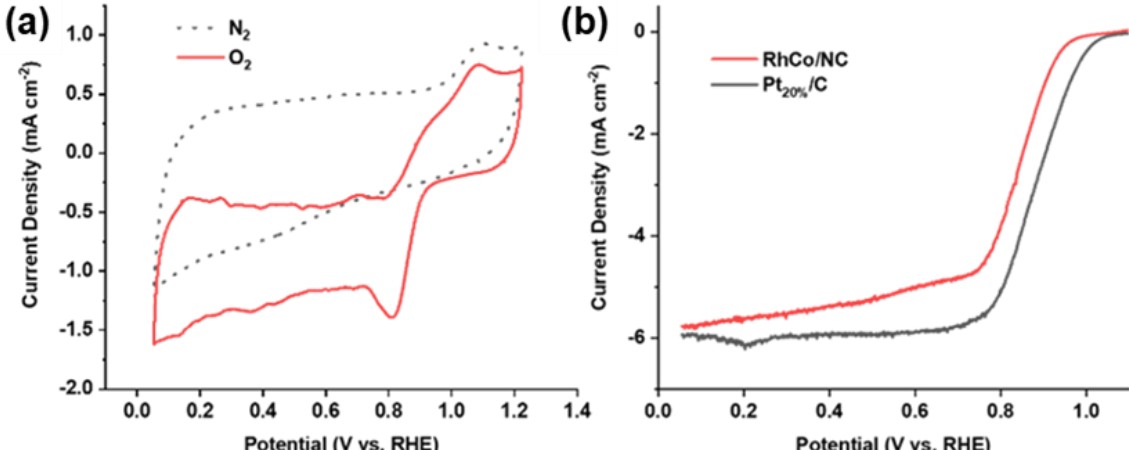

**Figure 2.** The CV curves of RhCo/NC in $N_2$ and $O_2$—saturated 0.1 M KOH solution; (**b**) the LSV curves of RhCo/NC and Pt/C in the $O_2$—saturated 0.1 M KOH.

Figure 3a shows the LSV curves of the RhCo/NC nanocatalyst at rotation speeds of 400–2400 rpm, and Figure 3b shows the K–L plots obtained at potentials of 0.3–0.6 V. These were used to determine the electron transfer number. Typically, the two-electron and four-electron pathways are the major mechanisms of the ORR. The two-electron pathway produces hydrogen peroxide as an intermediate product, and the four-electron

pathway generates water. Hydrogen peroxide typically reduces the catalytic activity by hindering the adsorption of a substrate onto catalytic sites. Thus, the four-electron pathway is most efficient mechanism of the ORR. As shown in Figure 3b, the K–L plots obtained at various potentials exhibited good parallelism. The electron transfer number for the ORR determined from the K–L slopes was 4.15. Thus, we verified that the RhCo/NC nanocatalyst promoted the four-electron pathway. The electrochemical active surface area was determined by calculating the double-layer capacitance ($C_{dl}$). The values of $C_{dl}$ for the RhCo/NC nanocatalyst (Figure S6) and Pt/C catalyst (Figure S7) were 38 mF cm$^{-2}$ and 44 mF cm$^{-2}$, respectively.

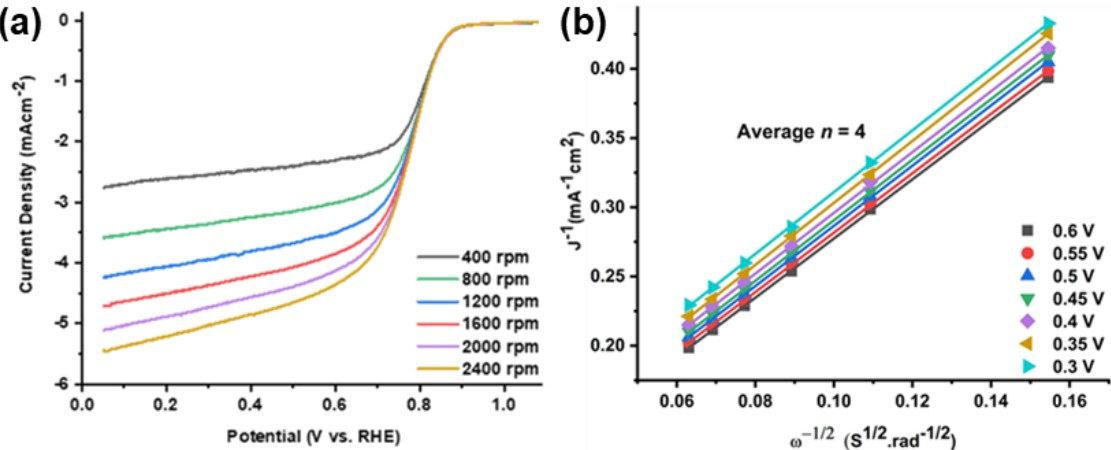

**Figure 3.** (**a**) The LSV polarization curves of RhCo/N at 5 mV s$^{-1}$ and various rotation speeds; (**b**) the K−L plots from 0.3 V to 0.6 V potential in $O_2$—saturated 0.1 M KOH electrolyte.

The long-term durability of the RhCo/NC nanocatalyst and Pt/C catalyst was investigated by performing chronoamperometry (CA) at 0.61 V vs. the RHE under the same conditions at a rotation speed of 1600 rpm. As shown in Figure 4a, the RhCo/NC nanocatalyst exhibited good durability, with 98% preservation of the initial current after 5000 s. However, the Pt/C catalyst showed a current loss of 93%. The outstanding structure of RhCo/NC increased the conductivity and effectively provided active sites. Thus, it exhibited stable catalytic activity; durability; and enhanced electron and mass transport. The methanol tolerance of the catalysts for the ORR is important in practical applications. Commercial Pt has poor durability owing to aggregation and the influence of toxic intermediates, such as carbon monoxide (CO) poisoning, on the Pt-based catalyst surface. Therefore, it is essential to develop catalysts that can replace Pt-based materials for ORR. As shown in Figure 4b, the durability against methanol crossover effect was measured via CA by adding 1.0 M methanol after 500 s at 0.61 V vs. the RHE in the O-saturated 0.1 M KOH electrolyte at a rotation speed of 1600 rpm. The results in Figure 4b revealed that the RhCo/NC nanocatalyst was superior to the Pt/C catalyst in terms of the methanol tolerance. The Pt/C catalyst was distinctly oxidized after the addition of methanol. However, the RhCo/NC catalyst maintained a stable initial current. In addition, its durability was similar to that obtained without methanol addition (Figure 4a). Furthermore, we confirmed the resistance to methanol from the polarization curves obtained in the O-saturated alkaline solution at a scan rate of 5 mV s$^{-1}$ and a rotation speed of 1600 rpm. Figure 4c showed a noticeable peak of methanol oxidation after the addition of methanol. Figure 4d was similar to the initial LSV curves. In addition, $E_{1/2}$ showed a negligible shift of 27 mV. Therefore, the ORR performance of the fabricated RhCo/NC nanocatalyst showed its potential as a catalyst.

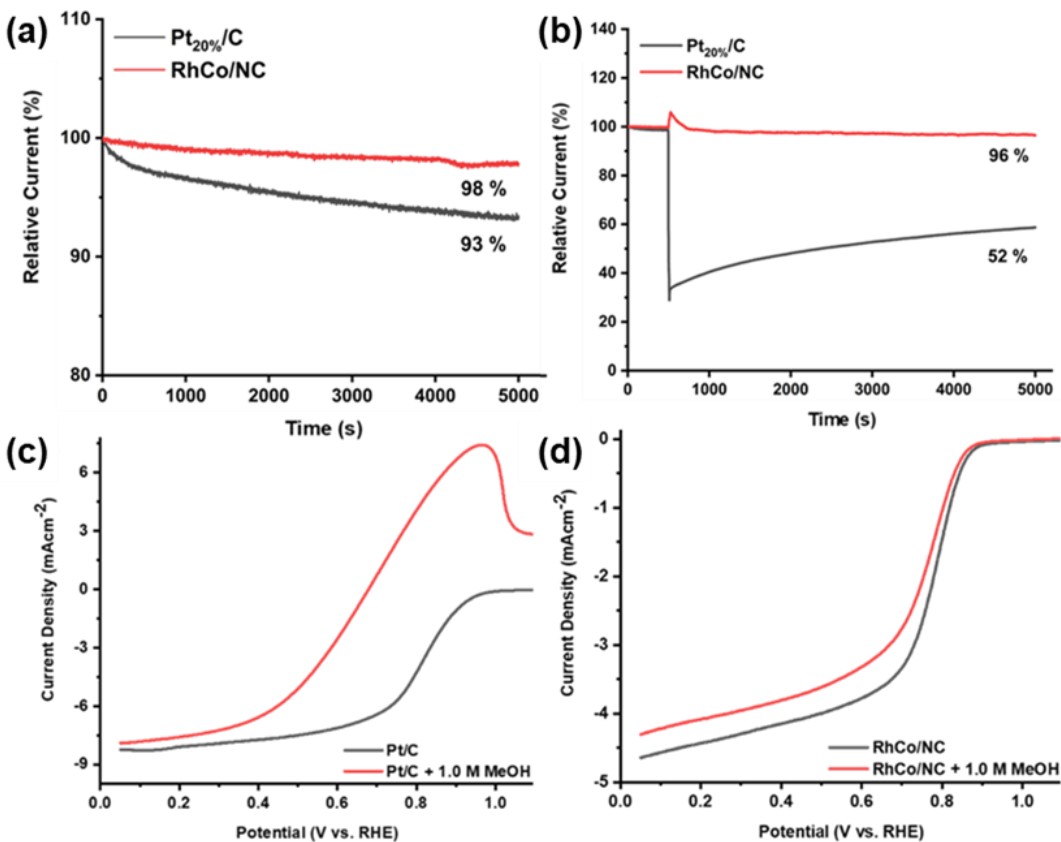

**Figure 4.** (**a**) The chronoamperometric curves of RhCo/NC and commercial Pt/C at 0.61 V using a rotation speed of 1600 rpm for 5000 s; (**b**) chronoamperometric curves after addition 1.0 M methanol at 500 s; polarization curves for (**c**) Pt/C and (**d**) RhCo/NC in the absence and presence of 1.0 M methanol in 0.1 M KOH electrolyte.

## 4. Conclusions

A facile synthesis method for a RhCo/NC nanocatalyst was developed. Nanosized Rh–Co alloy particles (6.2 nm) were uniformly dispersed in a N-doped C support at a considerably low loading mass. The nanocatalyst showed superior electrochemical activity for the ORR. The morphologies of the composites were characterized using TEM, XRD, XPS, and Raman spectroscopy. The electrocatalytic activity for the ORR was investigated using the RDE technique. The electrocatalytic activity with a better tolerance to methanol crossover and stable long-term durability was successfully identified by the edge effect and the synergistic effect of Rh–Co alloy nanoparticles and the N-doped C support. This nanocatalyst was cost-effective and highly durable, and it showed remarkable methanol tolerance. These results demonstrate the potential of using the RhCo/NC nanocatalyst as an alternative to commercial Pt-based catalysts. Moreover, the RhCo/NC catalyst is expected to function as a bifunctional electrocatalyst in the hydrogen evolution reaction and ORR.

**Supplementary Materials:** The following supporting information can be downloaded at https://www.mdpi.com/article/10.3390/pr10112357/s1, Characterizations; Figure S1: Histogram of particles size distribution of RhCo/NC; Figure S2: The XRD spectrum of RhCo/NC; Figure S3: The Raman spectrum of RhCo/NC; Figure S4: (a) XPS survey spectrum of RhCo/NC and XPS spectrum in the energy regions of (b) Rh 3d (c) Co 2p, and (d) N 1s; and Figure S5: Tafel curves of RhCo/NC and Pt/C; Figure S6: The CV curves of RhCo/NC in 0.1 M KOH solution at different scan rates, (b) linearly fitted curve of corresponding scan rate vs capacitive currents; Figure S7: The CV curves of Pt/C in 0.1 M KOH solution at different scan rates, (b) linearly fitted curve of corresponding scan rate vs capacitive currents.

**Author Contributions:** Conceptualization, S.P. (Sujung Park), N.P., and K.H.P.; methodology, S.P. (Sujung Park); software, S.P. (Sujung Park) and M.N.; validation, S.P. (Sujung Park), N.P., S.P. (Sungkyun Park) and K.H.P.; formal analysis, N.P. and M.N.; investigation, S.P. (Sungkyun Park); resources, K.H.P.; data curation, S.P. ( Sujung Park), N.P., and M.N; writing—original draft preparation, S.P. (Sujung Park), N.P. and M.N.; Revision, S.P. (Sujung Park), N.P., M.Y. and M.N.; visualization, S.P. (Sujung Park) and N.P.; supervision K.H.P.; project administration, K.H.P. and J.-M.L.; and funding acquisition, K.H.P. All authors have read and agreed to the published version of the manuscript.

**Funding:** This work was supported by the Materials/Parts Technology Development Program (20017575, the Development of the Applicability Evaluation Technology for Cryogenic Insulation Material and Storage Vessel considering the Operating Conditions of Hydrogen Commercial Vehicles) funded by the Ministry of Trade, Industry, and Energy (MOTIE, Korea) and the Basic Science Research Program through the National Research Foundation of Korea (NRF) grant funded by the Korea Government (NRF-2021M3I3A1084719).

**Institutional Review Board Statement:** Not applicable.

**Informed Consent Statement:** Not applicable.

**Data Availability Statement:** Not applicable.

**Conflicts of Interest:** The authors declare no conflict of interest.

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
