# Peer review of "Facile Synthesis of Nitrogen-Doped Carbon-Supported Rhodium–Cobalt Alloy Electrocatalyst for Oxygen Reduction Reaction"

_processes, doi:10.3390/pr10112357_

Round 1

Reviewer 1 Report

In this work a Facile Synthesis of Nitrogen-doped Carbon Supported  Rhodium Cobalt Alloy Electrocatalyst for Oxygen Reduction  Reaction is presented. In general the work is well written and the results are in accordance with the objectives of the manuscript. I recommend its publication after minor revision.

Insert and cite Figure 1 in the experimental section.

Discuss the stability of the material, can there be leaching?

What is the role of crystalline phases?

What is the specific area of the material, is it an important factor to consider?

Author Response

We sincerely thank the reviewers for their valuable comments and suggestions raised in our manuscript. We have modified the manuscript accordingly (correction texts are marked by red color) and detailed answers to the questions are given as follows

Reviewer 2 Report

The authors' work presents quite interesting data, but there are the following comments:

1. 200 nanoparticles is a small value for determining the average size of nanoparticles. Please increase the number of nanoparticles in processing.

2. Figure 2b shows the activity curves of catalysts in ORR.The magnitude of the current density depends on the surface area of the disk electrode and its rotation speed. Based on this, both curves should have the same values of the limiting diffusion current. Why do the values differ by 1 mA cm-2 at 0.6 V?

3. The stability of catalysts given with an accuracy of hundredths of % is unlikely to be reproducible. Round the values to integers. 

4. It is very interesting to give the ESA area values for the studied materials.

5. The authors write "The electrocatalytic activity with a high CO-like corrosion tolerance and stable long-term durability is successfully identified by the edge effect and the synergetic effect of Rh-Co alloy nanoparticles and nitrogen doped carbon supported". Give the cyclic voltammograms of CO oxidation in the article and comment on them accordingly.

Author Response

(The authors gave the same response as above.)

Reviewer 3 Report

In the article, we understand (as much as we can) that the authors have designed Rh-Co nanoparticles supported by nitrogen-doped carbon. The synthesis was performed by ball-milling. The catalytic activity was analyzed by various methods and compared with classical Pt/C.

However, the soundness and extent of this research are impossible to assess because of the language level. If a colleague had asked me to proof-read such a paper, I would have made so many changes and re-written so many sentences (most of the paper, so to speak) that I would have demanded that my name be among the authors.

In this article, the syntax is so appalling, the grammar so shocking that one can hardly understand anything. There are so many grammatical errors, malaproprisms (e.g. line 60: “They’re” for “their”), misuses of adverbs as adjectives (e.g. line 43: “highly efficiency”), approximations in the conjugation and nonsensical phrases of all sorts that they cannot be listed without copy-pasting the entire article.

It is of note that the frequent discontinuities in the expression of ideas and the wrong word orders often lead to logical incoherence. This is not acceptable in a scientific demonstration.

One cannot write a proper scientific article in their own language and just hope that literal translation can do the job without asking someone that correctly master English to help and write the paper. This communication is not written in English but in gibberish and can therefore most certainly not be published.

Author Response

(The authors gave the same response as above.)

Round 2

Reviewer 2 Report

Unfortunately, the authors did not take into account all the comments of the reviewer. The reviewer received convincing responses to the comments. However, I draw the attention of the authors of the study that the limiting diffusion current does not depend on the activity of the catalyst. The limiting current depends on the electrode area, depolarizer concentration, kinematic viscosity, etc. I ask the authors to pay attention to this fact when planning new research.

Author Response

We sincerely thank the reviewers for their valuable comments and suggestions raised in our manuscript. We have modified the manuscript accordingly and detailed answers to the questions are given as follows

Reviewer 2

Comment: Unfortunately, the authors did not take into account all the comments of the reviewer. The reviewer received convincing responses to the comments. However, I draw the attention of the authors of the study that the limiting diffusion current does not depend on the activity of the catalyst. The limiting current depends on the electrode area, depolarizer concentration, kinematic viscosity, etc. I ask the authors to pay attention to this fact when planning new research.

Responses: We sincerely thank the reviewer for their valuable time and suggestions raised on our response. The steady-state limiting current density (JL) indicates the maximum diffusion current density in the ORR during RDE measurement, which should be a fixed value in theory for a four-electron transferred ORR in a particular concentration solution and at a certain rotate speed. Many factors, including catalyst selectivity, electrode area, electrolyte viscosity, and O2 diffusion, could affect the measured JL based on the equation (Levich equation). However, in experiments, JL is always variable than theoretical value even though with the same the catalyst, electrode, and rotator (ChemElectroChem 2020, 7, 1107–1114). So, the impact of various experimental operating parameters on JL is highly necessary to be investigated. As per the Reviewer #2 suggestions we will pay more attention for our upcoming research.

The manuscript has been thoroughly revised, and submitted for the publication. We believe this manuscript has been substantially improved and we look forward to your positive response.

Best Regards

Kang Hyun Park

Reviewer 3 Report

Dear authors, 

I note a slight improvment of the article. On the scientific side, the article is probably correct, I understand the purpose of your work. But the language still needs extensive corrections. You will find below the first 100 lines that I tried to correct regarding the English style. Can you do the same work for the entire article

You have to add some references, for instance for the ORR reaction (line 17), for the Platinum (line 44)...

Can you explain CO-like corrosion? 

Desintegration is only used in the case of nuclear desintegration 

Fuel cell is considered an efficient and eco-friendly alternative 16

for energy production. One of the most important reactions is the oxygen reduction reaction (ORR) 17

 in energy converting systems for fuel cell. Herein, rhodium and cobalt alloy nanoparticles 18

 well-deposited on nitrogen-doped carbon supports (RhCo/NC) were synthesized by ball-milling 19

 and thermal decomposition methods. The RhCo/NC composites were transformed into nanopar- 20

 ticles with an average diameter of approximately 4 nm. The properties of the as-synthesized 21

 RhCo/NC nanocatalyst were characterized by transmission electron microscopy (TEM), X-ray dif- 22

 fraction (XRD), X-ray photoelectron spectroscopy (XPS), and Raman spectroscopy. The catalytic ac- 23

 tivity of the nanocatalyst was investigated toward ORR. In particular, the RhCo/NC nanocatalyst 24

 showed a good activity for ORR, a long-term durability in chronoamperometry (CA) tests and a re- 25

 sistance against methanol crossover in alkaline solutions, due to the synergistic effects of the metal 26

 alloy. Chronoamperometric analysis demonstrated the remarkable durability of the RhCo/NC 27

 nanocatalyst compared with commercial Pt/C. Moreover, RhCo/NC nanocatalyst shows good meth- 28

 anol tolerance property. In conclusion, the nanocatalyst developed in this work could offer an interesting alternative to platinum-based catalysts.

To face the current increasing energy consumption and overcome the problems caused by the toxic air pol- 35

 lutants, the novel energy systems have to be renewable and sustainable, by devel- 36

 oping green energy conversion and storage devices, such as fuel cells (FCs) like proton ex- 37

 change membrane fuel cells (PEMFCs) or direct methanol fuel cells (DMFCs), and metal- 38

 air batteries.[1–4] Fuel cells have in particular been spotlighted as eco- 39

 friendly energy devices because only water and waste heat are released while high energy conversion effi- 40

 ciency from chemical energy to electrical energy is obtained. To design FCs, the devel- 41

 opment of catalysts is vital at the cathode.[2,5–7] The oxygen reduction reaction 42

 (ORR) is slow and corresponds to the rate-determining step.[8,9] 43

Platinum is amongst the most efficient metal regarding the electrocatalytic activity[DE3] , particularly for ORR at the 44

 cathode in the fuel cells. Despite its qualities, platinum-based catalysts have intrinsic drawbacks such as 45

its prohibitive price, scarcity, low durability and weak methanol tolerance during ORR, which 46

have severely restricted their industrial applications. To solve the aforementioned issues, numerous 47

 attempts have been made for reasonable design and construction of platinum group elec- 48

trocatalysts by slightly modifying the size, shape, structure and compositions.[10,11] One 49

of the most successful strategies is the composition of alloy nanocatalysts with one of the crucial 50

metals (Ru, Pd, Ag, Rh, etc.)[10–14] and a non-noble metal (Co, Fe, Mn, Ni, etc.).[15–20] 51

Rhodium (Rh) is an interesting candidate for ORR catalysis, thanks to its versatility, 52

high activity and selectivity towards many electrochemical reactions.[21,22] Besides, Rh 53

has a high tolerance to CO-like  corrosion and a good stability under harsh conditions 54

as compared to other metals.Transition metals exhibit significantly 55

high ORR performance but their limitation to relatively low surface area and poor conductivity 56

seriously hampering their practical application. Such metals have therefore been integrated 57

into a nitrogen-doped carbon matrix to solve these issues.[23,24] 58

Herein, we report a facile preparation of nitrogen-doped carbon-supported rhodium- 59

cobalt alloy nanoparticles (RhCo/NC) by a high-energy ball-milling method and thermal 60

treatment of a solid mixture of metal salts and urea under nitrogen gas flow (Scheme 61

1). The ball-milling is commonly used in the synthesis of heteroatom-doped carbon supports. 62

Ball-milling is a well-known technique to mechanically produce uniform-sized particles 63

and the thermal treatment allows a larger surface area of the composites.[7,20,25] The 64

uniformly dispersed RhCo/NC nanocatalyst thus obtained has been utilized as electrocatalyst in65

ORR and its performance has been evaluated.

l. 74 : nitrogen-doped carbon-supported rhodium-cobalt alloy nanoparticles

l. 75 : molar amount ratio

The electrochemical performance was assed towards ORR by using a potentiostat (ZIVE SP1, Zi- 87

velab) and a modulated speed rotator (AFMSRCE, Pine) at room temperature. The three-electrode setup was composed of a satu- 88

rated calomel electrode (sat. NaCl) as reference , of a graphite electrode as 89

counter electrode and a glassy carbon rotating 90

disk electrode (RDE, diameter 5mm) as working electrode. In this article 91

, all the potentials were given against RHE reference, and the conversion from a SCE to a RHE scale was performed with the following equation: ???? = ???? + 0.0592 × ?? + ???? ° in 0.1 M KOH alkaline electrolyte92

. Both RhCo/NC and commercial 93

20 wt% Pt/C were dispersed for 1 h in 2-propanol (200 μL), deionized water (200 μL) and 94

5% Nafion solution (20 μL) to make homogeneous inks by sonically mixing 5 mg of the 95

catalyst. 7 μL of nanocatalyst ink was loaded on the surface of a rotating disk electrode 96

and dried at 60 ℃ for 1 h. All of the ORR performances were evaluated in 0.1 M KOH 97

alkaline electrolyte after purging with N2 or O2 gas for 30 min. The cyclic voltammetry 98

(CV) was performed in the 0.05-1.22 V range for 30 cycles at a scan rate of 10 mV s-1,the linear sweep voltammetry (LSV) conducted at a scan rate

of 5 mV s-1 with various rotation rates 100

from 400 rpm up to 2400 rpm, and the chronoamperometry carried out at 0.56 V with a rota- 101

tion rate of 1600 rpm. The long-term durability and the methanol tolerance were 102

investigated in 0.1 M KOH aqueous solution, and 1.0 M methanol was added after 500 s. The 103

number of electron transfer (n) is calculated from the Koutecky-Levich (K-L) plot, which 104

followed the equation below (1)-(3). 105

Author Response

Dear Ms. Freya Feng,
Special Issue Managing Editor,

We sincerely thank the reviewers for their valuable comments and suggestions raised in our manuscript. We have modified the manuscript accordingly and detailed answers to the questions are given as follows

The manuscript has been thoroughly revised, and submitted for the publication. We believe this manuscript has been substantially improved and we look forward to your positive response.

Best Regards

Kang Hyun Park

Round 3

Reviewer 3 Report

Dear authors, 

thank you for the corrected version and your answers 

I appreciate the editing work of your article. 

regards

Author Response

We sincerely thank you for your detailed reviews and suggestions to improve the quality of our manuscript.